



Meteorological factors driven glacial till changing and the associated
periglacial debris flows in Tianmo Valley, southeast Tibetan Plateau
Mingfeng Deng[1,2], Ningsheng Chen[1]*, and Mei Liu[1,2]
([1] Key Laboratory of Mountain Hazards and Surface Process, Institute of Mountain Hazards and Environment,
Chinese Academy of Sciences, Chengdu 610041, China;
[2] University of Chinese Academic of Sciences, Beijing 100049, China)
Abstract: Meteorological studies have indicated that high Alpines are strongly affected by climate
warming. Periglacial debris flows are more frequent in deglaciated regions. The combination of
rainfall and air temperature controls the initiation of periglacial debris flows; and the addition of
melt-water due to higher air temperatures enhances the complexity of the triggering mechanism
compared to storm-induced debris flows. In south-eastern Tibetan Plateau where temperate
glaciers are widely distributed, numerous periglacial debris flows have occurred in the past 100
years, but none had happened in the Tianmo watershed until 2007. In 2007 and 2010, three
large-scale debris flows occurred in the Tianmo watershed. In this research, these three debris flow
events were chosen to analyze the impact of the annual meteorological conditions: the antecedent
air temperature and meteorological triggers. TM images and field measurement of the nearby
glacier suggested that a sharp glacier retreat had existed in the previous one or two years
preceding the events, which coincided with the spiked annual air temperature. Besides, changing
of glacial tills driven by prolonged increase in the air temperature is the prerequisite of periglacial
debris flows. Triggers of periglacial debris flows are multiplied and they could be high intensity
rainfall as in DF1 and DF3, or continuous percolation of melt-water due to the long term rising air
temperatures as in DF2.
**1. Introduction**

24        The alpine environments are strongly vulnerable to climate changes, of which the

alpine glaciers and permafrost are the most sensitive in the form of glacier and
permafrost retreat (Harris et al, 2009; IPCC, 2013). Glacier and permafrost retreat can
induce mass movement, such as landslides, shallow slides, debris, moraine collapses,
etc. (Cruden and Hu, 1993; Korup, 2009; McColl, 2012; Stoffel and Huggel, 2012;
Fischer et al, 2012), that will be expelled out of the watershed in the form of debris
flows or sediment flux. The debris flow in alpine areas can often bury residential areas,
cut off main roads and block rivers (Shang et al, 2003; Cheng et al, 2005; Deng et al,





2013) and destroy basic facilities located in downstream, posing a great threat to the
local economy and social development. In undeveloped alpine areas such as the
south-eastern Tibet where the traffic/drainage system is particularly poor or limited,
the negative effects produced by debris flows such as cutting off main roads are
serious (Cheng et al, 2005).

Periglacial debris flows characterize the high alpine areas containing large areas

of glaciers, such as the Tibetan Plateau in China(Shang et al, 2003; Ge et al, 2014),
Alps in Europe(Sattler et al, 2011; Stoffel and Huggel,2012), Caucasus Mountains in
Russia(Evans et al, 2009) and northern Canada(Lewkowicz1 and Harris, 2005).
Periglacial debris flows were reported to be initiated by rainfall (Stoffel et al, 2011;
Schneuwly-Bollschweiler and Stoffel, 2012), melt-water flow of glacier or ice particle
ablation(Arenson and Springman, 2005; Decaulne et al, 2005), or outburst floods
from glacier lakes (Chiarle et al, 2007) in different parts of the world, while the
multi-triggers for the case is rarely to be read. Because debris flows are commonly
triggered by rainfall (Sassa and Wang, 2005; Decaulne et al, 2007; Kean et al, 2013;
Takahashi, 2014), the rainfall threshold, intensity and duration has been widely used
for debris flow monitoring and giving warning in non-glacier areas (Guzzetti et al,

2008).

In deglaciation areas, the debris flow threshold can be more difficult to determine.

Periglacial debris flows tend to occur in the summer when the thawing of glaciers and
glacial tills predominates and melt-water penetrates into the glacial tills at a constant
and successive flow. The effect of melt-water appears similar to that of antecedent
rainfall (Rahardjo et al, 2008) and is variable in different periods, considering snow
and glacier shrinkage and air temperature fluctuation. In the Swiss Alps, melt-water is
high in early summer, and as debris flows can be initiated by low total rainstorm,
whereas higher total rainstorm are required in late summer or early autumn when the
melt-water is low (Stoffel et al, 2011; Schneuwly-Bollschweiler and Stoffel, 2012). In
south-eastern Tibetan Plateau, the rainfall threshold given by Chen et al., (2011) is
quite wide, and the small rainfall threshold in particular is likely to contain the effect
of air temperature. Moreover, periglacial debris flows induced by a sudden release of



water from dammed glaciers have a close relationship with the rising air temperature
(Liu et al, 2014).
Fluctuation of air temperature is likely to be quite important in triggering
periglacial debris flows. Compared with the storm induced debris flows, the addition
of air temperature can greatly enhance the complexity of the initiation of periglacial
debris flows. It is of high difficulty to simulate the triggering process by experiment
or mathematical simulation, and instead, debris flows cases in the natural environment
could be the perfect object. In this research, three debris flow events, after a
debris-flow-free period of nearly 100-year, in the Tianmo watershed of the
southeastern of the Tibetan Plateau as deglaciation continued are used as examples,
and the annual meteorological conditions, antecedent air temperature and triggering
conditions prior to debris flows are analyzed to further understand the meteorological
triggers and their roles in glacier retreat, glacial till change and debris flow initiation.

## 75  2. Background

### 76  (1) Study area

The temperate glacier in the Tibetan Plateau is primarily distributed in the
Parlung Zangbo Basin and covered a total landmass of 2381.47 $km^2$ in 2010 based on
TM images (Liu, 2013). Historically, the movement of temperate glacier has produced
a large amount of moraines, the depth of which can reach up to 500 m locally (Yuan et
al, 2007). In recent decades, there has been a dynamic significant increase in
temperature and according to statistics the temperature at the Bomi meteorological
station (midstream in the Parlung Zangbo Basin) has rose by 0.23°C/10a from 1969 to
2007, resulting in remarkable shrinkage of the glacier(Yang et al, 2010).
Tianmo Valley, located in Bomi County and to the south of the Parlung Zangbo
River, covers an area of 17.76 $km^2$ (29°59'N/95°19'E; Figure 1). This valley has a
northeast-southeast orientation and is surrounded by high mountains reaching 5590 m
a.s.l. at the southernmost site and 2460 m a.s.l. at the junction of the Parlung Zangbo
River. The TM image in 2013 showed the presence of a hanging glacier with an area



of 1.42 km$^2$ in the upper concave area at an altitude of 4246 m to 4934 m. Bared rock,
dipping at an angle of around 60°, emerged below and above the hanging glacier and
often covered by everlasting snow. Below 3800m a.s.l., vegetation, including forest
and shrub, occupies most of the area (Table 1).
The river channel in the watershed is sheltered by shade and not directly affected
by sunlight, resulting in less solar radiation and a location at which a small trough
glacier can form. In the main channel, the trough glacier extended to 2966 m a.s.l. in
2006. The lower part of the trough glacier has been eroded by glacier melt-water flow,
and an arch glacier that is vulnerable to high pressure was formed (Figure 2). The
remnants of the landslide deposits approximately 10 meters high, which consist of low
stability sediment and can be easily entrained by debris flows, can be observed in both
sides of the channel.
Tianmo Valley is on the north side of the bend in the Yarlung Zangbo River and
is strongly affected by the new tectonic movement. An inferred normal fault vertical
to the channel cuts through the valley and is only 30 km away from the Yarlung
Zangbo fault. In 1950, a rather significant earthquake (Ms. 8.6) hit Zayu, which is
only 200 km away, and local records reported that a large amount of rock collapsed
and landslides were produced at that time. The whole valley is in a strong ductile
deformation zone and is dominated by gneissic lithology belonging to Presinian
System.

**(2) Disaster history**

According to our field interview with local residents, there were no debris flows
in approximately 100 years prior to 2007 in Tianmo Valley. The channel was quite
narrow before 2007, and the local people could walk across via a wooden bridge to
live and farm on the terrace on the west side. The ecology was in a rather peaceful
state at that time.
On the morning of Sep. 4$^{th}$, 2007, after the rainfall which did not hit the
downstream area ceased, the local forest guard heard a loud noise coming from the
upstream area at approximately 18:00; with rainfall which later began in the upstream



area at approximately 19:00, following this rainfall was debris flows which rushed out
of the Tianmo Channel and subsequently blocked the Parlung Zangbo River; report
stated that several debris flows occurred, lasting the entire night. According to the
field measurements, approximately 1,340,000 $m^3$ of sediment was transported during
this event, resulting in 8 missing persons and deaths. Concurrently within this same
time, debris flows occurred in the four nearby valleys (Table 2). According to the size
classification proposed by Jakob (2005), which is based on the total volume, peak
discharge and inundated area, Size class of debris flows in the five valleys is given in
Table 2.

At 11:30 on Jul. 25$^{th}$, 2010, debris flows were again triggered in Tianmo Valley

that traced the path of the preceding debris flow deposits and reached the other side of
the Parlung Zangbo River. According to Ge et al., (2014), solid mass sediment of
approximately 500,000 $m^3$ was carried out (Table 1) and deposited on the cone to
block the main river. A barrier lake was formed, and the rising water destroyed the
roadbed of G318. The following week also experienced dozens of debris flows in
small magnitude.

Debris flows occurred again two months later on Sep. 6$^{th}$ (The Ministry of Land

and Resources P. R. C., 2010), although we could not determine the exact times
sequence of event but according to speculation, these debris flows could have
occurred in the early morning before dawn and when the rainfall intensity has reached
its maximum(Figure 9), which agrees with the findings of Chen (1991) that periglacial
debris flows have historically occurred between 18:00~24:00 in this area. The debris
barrier in the main river was consequently increased by an additional 450,000 $m^3$, and
the barrier lake was enlarged to maintain 9,000,000 m³ of water.

A field investigation revealed that a high percentage of boulders in the

downstream area and glacial tills above the trough glacier were quite loose and of
high porosity (Figure 2), hence they have low density and can be easily entrained. Our
particle size tests on the glacial tills and debris flow deposits indicate a lower clay
(d<0.005 mm) content, whereas the debris flow deposits contain more fine particles
that are smaller than 10 mm (Figure 4), suggesting that the entrainment supplied a





considerable amount of fine particles.

**(3) Meteorological data**

The study area is located in a high alpine area where the economy is quite
undeveloped with only few meteorological stations. Before 2011, the Bomi
meteorological station was the only station in the area, located 54 km away from
Tianmo valley at an altitude of 2730 m, and other stations were located more than 200
km away.
The Tibetan Plateau is a massive terrace that obstructs the Indian monsoon,
causing it to travel through the Yarlung Zangbo Canyon and its tributaries. As the
Indian monsoon is transported to higher altitudes, a rainfall gradient emerges in the
Parlung Zangbo Basin. However, according to our statistics on rainfall data in the area,
the rainfall is more or less the same from Guxiang to Songzong considering the
long-term rainfall process; therefore, the rainfall data from the Bomi meteorological
station can be used for our study. In order to conduct further study, another
meteorological station was built in 2011 near Tianmo Valley.
It has been established that the air temperature decreases with altitude; therefore
the air temperature in the source area of Tianmo Valley is lower than that in Bomi
County. According to the research by Li and Xie (2006), the air temperature decreases
at a rate of $(0.46\sim0.69)°C/100m$ over the whole Tibetan Plateau, and the rate in the
study area is $0.54°C/100$ m. Because the glacier and permafrost in the source area
have a planar distribution, the air temperature at the geometric centre of the glacier
and permafrost can be used to analyze the temperature process.

**3. Analysis and results**

**(1) Changing of air temperature and rainfall**

The annual air temperature is usually used to reflect the tendency of glacier
change (Yang et al, 2015). We collected annual air temperature and annual rainfall
data from 1970 to 2014 from the Bomi meteorological station (Figure 5). The record



showed that the overall air temperature has increased by approximately 1.5°C in the last 45 years, accounting for 0.033°C/a. This air temperature increase was particularly more rapid between 2005~2007, an approximately 0.7°C/3a, which is 7 times the average value of the last 45 years. On the other hand, the annual rainfall from 2000 to 2010 was low and it was estimated at 828.2 mm per year. From 2000 to 2004, the rainfall during summer (July to September) accounted for approximately 50% of the total annual rainfall; however, only 32% of the rainfall occurred in the summer of 2005~2006, even though the annual rainfall exhibited the same trend. In 2007, the rainfall in the summer and the entire year returned to normal.

According to Figure 5, a similar trend in the air temperature and rainfall was observed before DF2 and DF3. The air temperature elevated in 2009 to reach the maximum of the last 45 year period, accounting for 10.2 °C; however, the annual rainfall, was only 65% of the average amount; and the summer rainfall, lower than that in 2005 and 2006, reached their minimum values. In 2010, the rainfall was abundant and the annual rainfall increased to 1080.6 mm, which is approximately 30% more than the average value and close to the maximum.

The following common traits can be identified from comparing the annual meteorological conditions of DF1, DF2 and DF3. 1) One or two years before the debris flows, the annual temperature elevated and the annual rainfall and summer rainfall increased. The climate was in a "hot-dry" state. 2) As the temperature gradually decreased, the annual rainfall returned to normal or increased, and the "hot-wet" climate contributed to debris flow initiation (Lu and Li, 1989).

**(2) Changing of glacier in Tianmo valley**

In our research, remote image is collected to analyze the changing of glacier in the source area during the past years. In order to eliminate the effect of snow cover, images were taken in the thawing seasons when the snow cover is limited to enable an easy detection of the glacier from snow. Besides, a bright cloud is still needed to show the watershed clearly; however a difficult case ensues when the rainy season comes in-between the thawing season when the atmosphere is often covered by thick cloud.


Further, in order to show glacier retreat and its impact on debris flows properly, the
images should be within similar time interval, like 3 years, before and after debris
flow events. As the high resolution images are rare to obtain and we could only collect
one SPOT in 2008. To achieve consistency of the images, we collected 5 TM images,
taken on Sep. 17th, 2000, Jul. 24th, 2003, Sep. 21st, 2006, Sep. 24th, 2009 and Aug. 4th,
2013, respectively.

Based on the 5 TM images, we classified the area as glacier, snow, bared land,

gully deposition and vegetation in time series (Figure 6), and the area of each is given
in Table 1. Figure 6 showed that deglaciation was taking place in Tianmo valley and
in particular, the eastern branch had experienced the sharpest deglaciation. In order to
show clearly the rapid rate of glacier retreat, a graph was plotted to show the changing
of glacier and the eastern branch in Figure 7.

Figure 7 shows that glacier in Tianmo valley had been in shrinkage since 2000 to

2013, with variation in glacier retreat rate. In 2000~2003, 2003~2006, 2006~2009 and
2009~2013, the glacier retreat rate in Tianmo valley corresponds to 0.02, 0.06, 0.027
and $0.0075 km^2/a$ and 0.0033, 0.01, 0.008 and 0.002 $km^2/a$ for the eastern branch.
According to these figures the largest glacier retreat rate was in 2003~2006, followed
by that in 2006~2009. It is important that glacier area at the beginning should be taken
into consideration to judge the changing rate of glacier. The glacier retreat rate is
normalized and the relative glacier retreat rate is defined as:
$$D = \frac{(A_0 - A_1)}{nA_0} \qquad (1)$$

Where $D$ is the relative glacier retreat rate, $km^2/a/km^2$; $A_0$ is glacier area at the beginning,
$km^2$; $A_1$ is glacier area at the end, $km^2$; $n$ is the duration of year, a.

The relative glacier retreat rate are 11.30, 35.09, 17.43 and 5.17 $10^{-3} km^2/a/km^2$

during 2000~2003, 2003~2006, 2006~2009 and 2009~2013, respectively; whereas, it
is 20.83, 66.67, 66.67 and 20.83 $10^{-3} km^2/a/km^2$ for the eastern branch. These figures
show that the relative glacier retreat rate for the eastern branch had shrunk much more
sharply between 2000 ~2013.





In this research, TM images with 3 year intervals were applied can only get the

mean glacier retreat rate. As glacier retreat rate in the 3 three years could be either

high or low, field measurement of the nearby glacier is used to show the glacier retreat

condition before debris flows. Yang et al.(2015) had conducted field measurement of

No.94 Glacier in Parlung Zangbo Basin since 2006 and the field measurement

suggests it was in negative balance in 2006~2010(Figure 7). The negative balance

reached the maximal in 2009, followed by 2008 and 2006, indicating sharp

deglaciation in these three years.

When we combined the result of TM image and filed measurement of No. 94

Glacier, we observed that it is right before debris flows that glacier in Tianmo valley

experienced the sharpest deglaciation in 2006, 2008 and 2009, which was also

coincidental with the elevated annual air temperature (Figure 5). Besides, the

maximum glacier retreat in 2009 could be also related to the decline of snowfall in the

preceding winter and early spring and its increase may also have aided the glacier

retreat in 2007 and 2010.

## (3) Antecedent air temperature and rainfall process

The air temperature in the source area can be obtained using the vertical decline

rate (0.54°C/100 m). According to this method, the air temperature in the source area

was 9.8°C lower than that at the Bomi meteorological station. We collected the daily

temperature; that is the lowest temperature, the mean temperature and daily rainfall

from June to September in 2007 and 2010 (Figure 8).

According to Figure 8, the lowest air temperature was below 0 at the end of June,

2007. At the beginning of July, the air temperature started to rise quickly which

continued until early September when DF1 occurred, this demonstrates that the high

air temperature in July and August contributed to DF1.

According to Figure 8, the air temperature was high from early July to late

August, and another high air temperature period emerged in early September. When

DF2 occurred in late July the air temperature had reached the maximum for that year,

which suggests that the air temperature in early and middle July was responsible for





DF2. After DF2 occurred, the air temperature in August began to prepare for DF3.

Antecedent air temperature fluctuation includes the air temperature and its

duration. The air temperature and duration before debris flows are variable, making
them difficult to evaluate. The accumulation of positive air temperature is usually
applied to analyze the impact of air temperature on glacier melting (Rango and
Martinec, 1995), which can be expressed as:
$$T_{PT} = \sum_{i=-n}^{0} T_i (T_i > 0) \qquad (2)$$

Where $T_{PT}$ is the positive air temperature accumulation, °C and $T_i$ is the

average daily air temperature; only $T_i > 0$ is included.

Because air temperature is successive, it is difficult to determine the beginning of

positive air temperature accumulation. Glacial tills can lessen the heat that penetrates
into them, and the low air temperature can only contribute to the upper thin layer;
moreover, freeze-thaw cycles exist when the lowest air temperature is less than 0°C.
From this point of view, the beginning of positive air temperature accumulation is
defined as the time at which the lowest air temperature exceeds 0°C for several
successive days or the last debris flow.

Based on the above method, we can deduce that the positive air temperature

accumulation began when the lowest air temperature exceeded 0°C for several
successive days, starting on June 28[th], 2007 and June 9[th], 2010 corresponding to DF1
and DF2, respectively, and on July 26[th], 2010 for DF3, following DF2. The duration
and $T_{PT}$ were calculated for each debris flow event , the result was 69 days and
517.9°C, 47 days and 332.1°C, 42 days and 320.4°C (Figure 8) for DF1, DF2, and
DF3, respectively. The result showed that $T_{PT}$ for DF1 is much larger than the other
two, and the reasons for this may lie in the watershed there had been no debris flows
in the past dozens of years and only extraordinary external forces could have
destroyed the long-term balance.





## (4) Triggering conditions

The continuous nature of the air temperature limits the possibility for debris flows triggered by a sole abrupt increase in air temperature; and since the previous air temperature trend cannot be neglected, it is of no sense to study air temperature triggers.

Antecedent rainfall is a factor that favours debris flows. In our analysis, the rainfall over the three days preceding a debris flow event is given in Figure 9.

Before DF1, the air temperature was high, and continued through July and August. The $T_{PT}$ reached 517.9°C. According to the local forest guard, an isolated convective storm occurred prior to DF1 though no rainfall was recorded at the Bomi meteorological station or in the downstream area at that time. In Figure 9, as the rainfall right before DF1 occurred was not recorded by Bomi metrological station, we added to the rainfall intensity (like 5 mm/h) before DF1 to account for the storm, which does not reflect the rainfall during storm conditions. We can therefore conclude that this isolated convective storm initiated DF1, while the long-term high air temperature trend had paved the road for DF1. Considering a large deglaciated area, several other periglacial debris flows simultaneously also occurred near Tianmo Valley (Deng et al, 2013), which suggests the advantageous meteorological conditions for debris flow initiation.

DF2 took place when the air temperature reached the peak in 2010. The thaw season began in the middle of June, and the $T_{PT}$ reached 332.1°C. On July 24[th], one day before DF2, the air temperature reached the maximum value for that year. The rainfall record at the Bomi meteorological station shows that there had been no rainfall several days preceding DF2, and the local citizens also did not observe any rain either. The trigger of DF2 was likely the continuous percolation of melt-water due to the long term rising air temperature.

According to field interviews, several debris flows of small magnitude had also occurred before DF3. The air temperature decreased in late August but increased to



another high peak before DF3, and the $T_{PT}$ reached 320.4°C. Rainfall began 2 days
prior to DF3 and was steady the entire day before DF3. According to the rainfall trend
at the Bomi meteorological station, the rapid increase in rainfall intensity started 4
hours before DF3 and reached 3.8 mm/h, which was responsible for the initiation of
DF3.

## 4. Discussion

Debris flows initiation is the process when water source provokes the movement
of soil mass. In this research, we found that the three debris flows were triggered by
high air temperature and rainfall in DF1, high air temperature in DF2, and rainfall in
DF3 respectively. When we analyzed the date and the triggers for these events,
various questions came to mind that gave reasons to doubts: 1) Why debris flows did
not occur in 2006 or 2009 when deglaciation reached its peak and more ice melt water
was present; 2) Why DF1 and DF3 occurred in September when the air temperature
and the ice melt water was decreasing; 3) Why was there is no large scale debris flows
triggered by the previous heavier storm. It makes us believe that the impact of water
source on the magnitude and frequency of debris flows is quite low, or there could be
much more debris flows; and instead, soil source, including its magnitude and activity,
should be the predominate controller, just as Jakob et al., (2005) pointed out that the
recharge of channel should be the prerequisite for debris flows. However, in most
situations we cannot reach the source area to detect the soil source and the high-tech
remote sensing can just distinguish the scope of soil source. In the preriglacial area
where the glacial till is often covered by glacier or everlasting snow, changing of soil
source seems to be of high difficulty to detect. In this research, we try to combine the
meteorological condition and the literatures to discuss the probable change of glacial
tills before debris flows.

### (1) Changing of glacial till in annual years

Climate warming is a global trend (IPCC, 2013), and the Tibetan Plateau, as the
third pole, is no exception. According to our statistics, the air temperature in Bomi



County has increased by 1.5° in the last 45 years (1970~2014). Glacier retreat induced
by climate warming has been widely accepted, and recent research suggests the
weaker Indian monsoon could be another reason ( Yao et al,2012). Glaciers are
always located in concave ground and cover a large amount of glacial tills.
Gravitation of the glacier can generate normal stress vertical to the slope, which can
strengthen the slope stability. The effect of glaciers on slope stability is called glacial
debuttressing (Cossart et al, 2008). As deglaciation continues, the result could lead to
exposure of the frozen glacial tills (Figure 10, A to B) and smaller glacial
debuttressing.

The retreat of glaciers and glacial tills with climate warming is quite different.

Deglaciation is accompanied by melting of internal ice particles. The internal ice
particles are covered by active glacial till and though the effect of heat fluxes are
strong at the surface and quite limited in deep layers, resulting into the melting of
internal ice particles lagging behind glacial retreat (Hagg et al, 2008). Glacial till with
thicker coverage has a relatively thinner thawing layer, although the ablation rate of
glaciers and internal ice particles remain at the same pace at the junction with the
slope. Newly formed bared glacial till is of high ice content and frozen, the cohesion
of the ice particles renders the bared glacial till with high shearing strength and
stability. Therefore, we often see many bare moraine slopes near glaciers, for this
reason there were no debris flows of large magnitude in 2006 and 2009 when glacier
retreat reached the maximal.
**(2) Changing of glacial till in antecedent days**

After the long term cold winter, the whole glacial tills would become frozen. If

the regressive glacier was not recovered in the winter, the glacial tills would often be
covered by snow. As air temperature increases again, the surface snow would melt
first, followed by the internal ice particles. The thawing of internal ice particles would
induces a series of changes in the glacial till, which include the following: 1) the
thawing will break the bonds of ice particles and increase the instability between ice
cracks (Ryzhkin and Petrenko, 1997; Davies et al, 2001); 2) the sharp air temperature

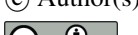


fluctuation in high alpine mountainous areas induces a repeated cycle of expansion
and contraction in the glacial till that can destroy the mass structure to some extent; 3)
the seepage of ice melt-water can deliver fine-grained sediments that were formerly
frozen in the ice matrix (Rist, 2007); and 4) the ice melt-water can result in a higher
water content and pore water pressure (Christian et al, 2012). These changes in glacial
till can sharply decline the soil strength, shifting to an active mass from the uncovered
and frozen moraine (Figure 10, B to C). Because the heat conduction in glacial till is
quite slow, this process may last for a very long time and also requires a high
antecedent air temperature.

Heat conduction via the percolation of rainfall and ice melt-water can amplify

the scope of an active of glacial till (Gruber and Haeberli, 2007), whereas the shelter
of surface glacial till can hinder the heat flux from the internal mass. At a low air
temperature, the heat flux should be constrained to the surface layer, and a large heat
gradient due to a high air temperature would contribute much more to the heat flux
and ice melt in the deep mass, meaning that the long-term effect of a high air
temperature can amplify the active glacial till (Åkerman et al, 2008), under which lies
frozen glacial till with a high ice content. The activity of glacial till changes with
depth, high in the surface and low in the deep layers, and landslide failure can take
place on glacial till slopes in a stepwise manner, coinciding with long-term air
temperature fluctuations although the glacial till is significantly unlimited in
deglaciation areas.
**(3) Failure of glacial tills**

Active glacial till slopes with low strength are usually vulnerable, and their

failure can occur when the air temperature is above $0^o$C (Arenson and Springman,
2005). Either rainfall, the seepage flow of glacier or ice particle melt-water induced
by prolonged high air temperature could trigger the failure (Figure10, C to D). The
failure mechanism lies in the ablation of internal ice particles and the percolation of
melt-water that further decreases the soil strength at first (Arenson and Springman,
2005; Decaulne et al, 2005); later, the subsequent rapid percolation of melt-water or





rainfall can saturate the glacial till and initiate failure through the decrease of soil
suction and shearing strength (Springman et al, 2003; Decaulne and Sæmundsson,
2007; Chiarle et al, 2007).
The fluctuation of air temperature within a specific low range result into limited
seepage flow. Based on the hypothesis that the glacier is limited, it is unlikely for
failure to be triggered by short-term increases in air temperature; although prolong air
temperature increases can still trigger it. Rainfall can initiate debris flows from active
glacial tills with a mechanism similar to that of storm-induced debris flows in
non-glacier areas (Iverson et al, 1997; Springman et al, 2003; Sassa and Wang, 2005).
In the European Alps, periglacial debris flows are mainly provoked by rainfall, which
is also related with air temperature fluxes (Stoffel et al, 2011). The different portion
containing melt-water percolation would impact the rainfall intensity and duration
required for periglacial debris flows (Stoffel et al, 2011; Schneuwly-Bollschweiler and
Stoffel, 2012); Rainfall intensity and duration may also require other preconditions,
such as the distribution of glaciers and frozen glacial tills and the terrain of the source
area to enhance the debris flow (Lewkowicz and Harris, 2005).
The three debris flow events possess similar annual meteorological conditions,
except that the positive air temperature accumulation prior to DF1 was significantly
larger. DF1 occurred at the end of a prolonged period of high air temperature, prior to
this, there were instances of failure but no large-scale debris flows. On July 25[th] 2010
when the daily rainfall particularly reached 20.7 mm, no debris flows were generated
because thick active glacial till was still lacking after small failure events. In 2010, the
largest daily rainfall occurred on June 7[th], accounting for 37.5 mm, at the beginning of
an air temperature increase when the glacial till was frozen and had low activity. The
lack of glacial till activity was the likely cause of the absence of debris flows. On
August 23[rd], the daily rainfall was 20.3 mm, the antecedent air temperature
accumulation dated from DF2, and the active glacial till was still under development.
On September 6[th], the antecedent positive air temperature accumulation was smaller,
and a low air temperature had emerged previously; however, the high rainfall intensity
supplemented this lack of prolonged high air temperature.



## 5. Conclusion

Climate changes have serious effects on high mountainous areas, and mass movement of sediments such as periglacial debris flows is increasingly frequent. Prolonged increases in the annual air temperature are regarded as very favourable for periglacial debris flows. In particular, the annual "hot-dry" weather condition one or two year earlier was responsible for the three debris flow events in Tianmo valley. Debris flow is usually not initiated in the first year because the melting of internal ice particles lags behind the glacial retreat result from the prolong air temperature rise.

Glacial till is unlimited in the deglaciated area, while its activity relies on glacial retreat and internal ice particle melting. Changing of glacial tills induced by increasing air temperature is the first step of periglacial debris flows and glacial till need a four phase experience prior to debris flow occurrence (these include:- glacier-covered glacial till, uncovered and frozen glacial till, active glacial till and debris flows), during which the varied air temperature condition with different factor drives the changing. The annual air temperature can remove glaciers, decrease glacial debuttressing and produce bared glacial till; the activity of the frozen glacial till is quite low and would be enhanced by prolonged high air temperature trends; active glacial till would fail and generate debris flows from multiple triggers, such as rainfall or the continuous percolation of ice melt-water. For periglacial debris flows of a large magnitude, the long-term effect of air temperature is required, although rainfall can shorten the antecedent period and generate debris flows earlier.

It is difficult to observe the changes of glacial till in source areas of debris flow, and the analysis of the phase conversion of glacial till in this research is based on the conditions that trigger debris flow and other literatures. Indeed, the meteorological conditions, such as the antecedent air temperature and meteorological triggers that drive the phase conversion are partly overlapped and difficult to distinguish. In the first study, we hope to distinguish the effect of each meteorological condition and more detail study should be done in further research.





**Acknowledgements**: This research was supported by the National Natural Science Foundation
of China (grant No. 41190084, 41402283 and 41371038) and the "135"project of IMHE, CAS. We
wish to acknowledge the editors in the Natural Hazards and Earth System Science Editorial Office
and the anonymous reviewers for constructive comments, which helped us in improving the
contents and presentation of the manuscript.

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






Table 1 Changing of glacier, snow, bared land, gully deposition and vegetation in Tianmo valley

| Year | Glacier (km²) | Glacier(eastern branch) (km²) | Snow (km²) | Bared land (km²) | Gully deposition (km²) | Vegetation (km²) |
|---|---|---|---|---|---|---|
| 2000 | 1.77 | 0.16 | 2.13 | 2.80 | 0.44 | 10.46 |
| 2003 | 1.71 | 0.15 | 2.44 | 2.54 | 0.44 | 10.48 |
| 2006 | 1.53 | 0.12 | 2.68 | 2.44 | 0.44 | 10.55 |
| 2009 | 1.45 | 0.096 | 2.81 | 3.03 | 0.47 | 9.90 |
| 2013 | 1.42 | 0.088 | 1.74 | 3.83 | 0.51 | 10.17 |


Table 2 Basic information of the debris flows in Tianmo and the nearby valleys

| No. | Name | Coordinates | Basin area (km²) | Glacier area (in 2006) (km²) | Date | Size class |
|---|---|---|---|---|---|---|
| 1 | Tianmo valley | 29°59'N 95°19'E | 17.74 | 1.53 | 4 Sep. 2007 | 6 |
| | | | | | 25 Jul. 2010 | 5 |
| | | | | | 6 Sep. 2010 | 5 |
| 2 | Kangbu valley | 30°16'N 94°48'E | 48.7 | 1.06 | 4 Sep. 2007 | 3 |
| 3 | Xuewa valley | 29°57'N 95°23'E | 33.22 | 0.95 | 4 Sep. 2007 | 2 |
| 4 | Baka valley | 29°53'N 95°33'E | 22.15 | 2.46 | 7 Sep. 2007 | 3 |
| 5 | Jiaqing Valley | 30°16'N 94°49'E | 15.51 | 1.12 | 9 Sep. 2007 | 3 |




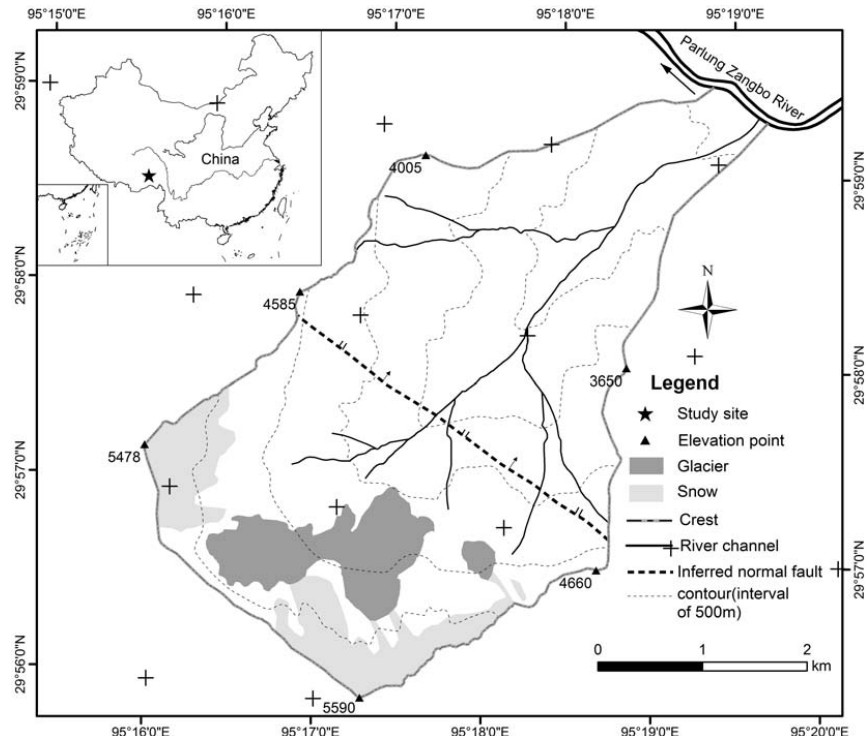

Figure 1 Location and basic information of Tianmo Valley

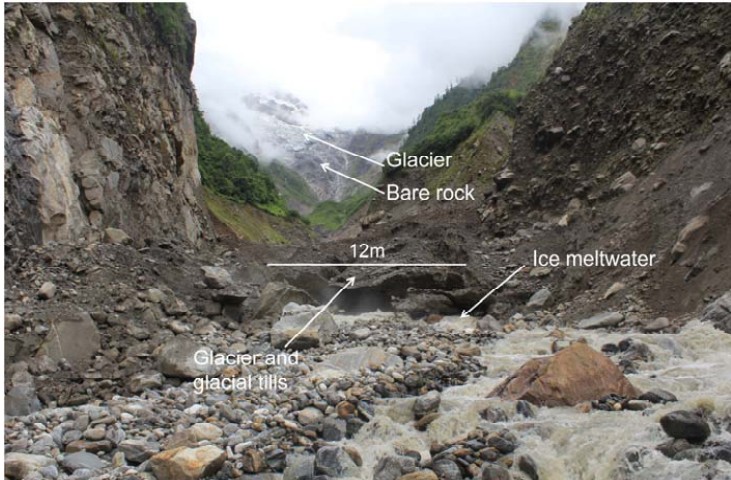

Figure 2 Overview of the valley from the channel(in 2014)




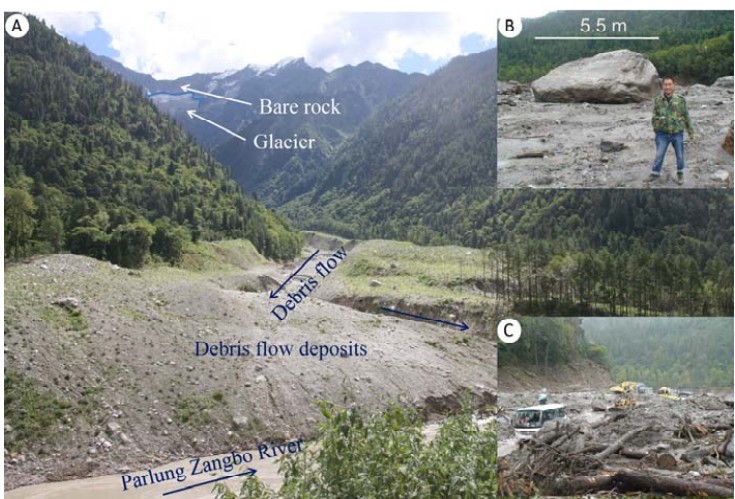


Figure 3 DF1 in 2007(A. Overview of Tianmo debris flows from the downstream area; B& C.

Boulder and debris flow deposits on the north side of the Parlung Zangbo River)

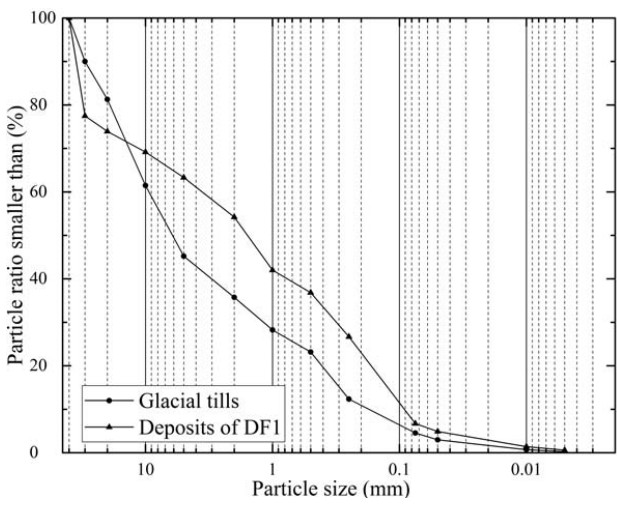


Figure 4 Particle size distributions of the glacial tills and debris flow deposits





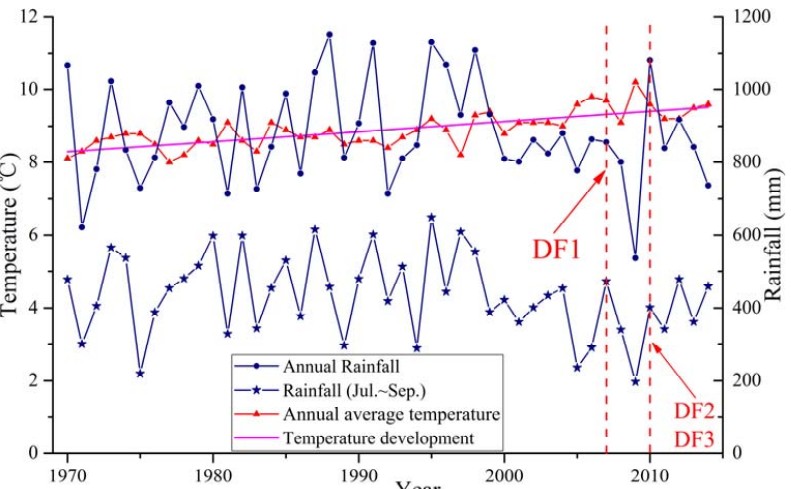


Figure 5 Variation of the annual air temperature and rainfall in Bomi, 1970 to 2014

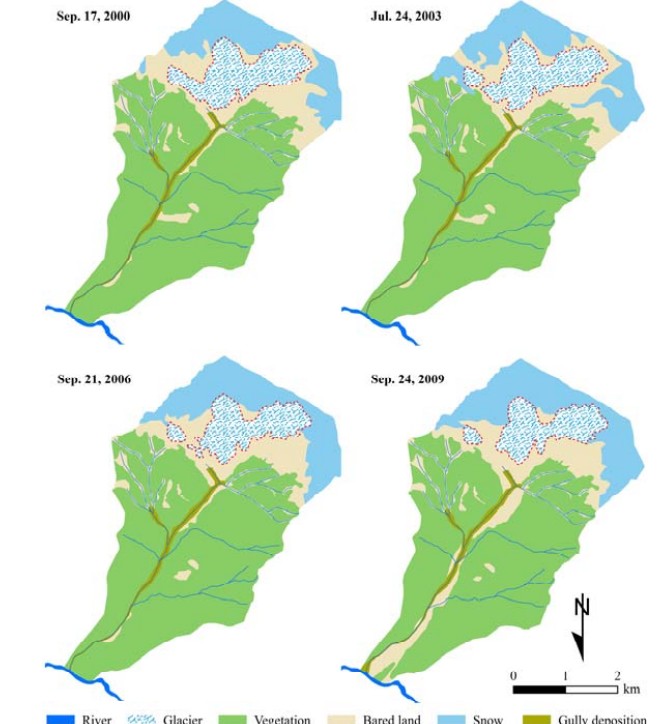


Figure 6 Distribution and changing of glacier, snow, bared land, gully deposition and vegetation in

Tianmo valley





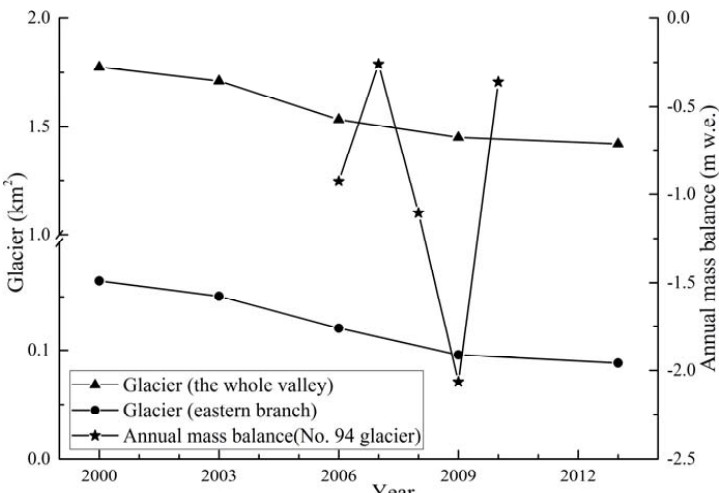


Figure 7 Changing of glacier via time and the measured annual mass balance for the

Parlung No. 94 Glacier (mass balance is edited by Yang et al.(2015))

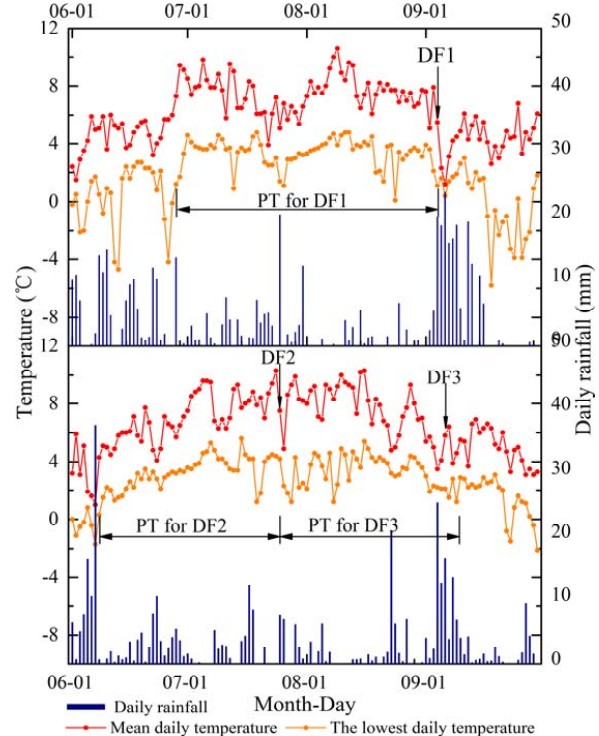


Figure 8 Air temperature and rainfall before and after DF1, DF2 and DF3


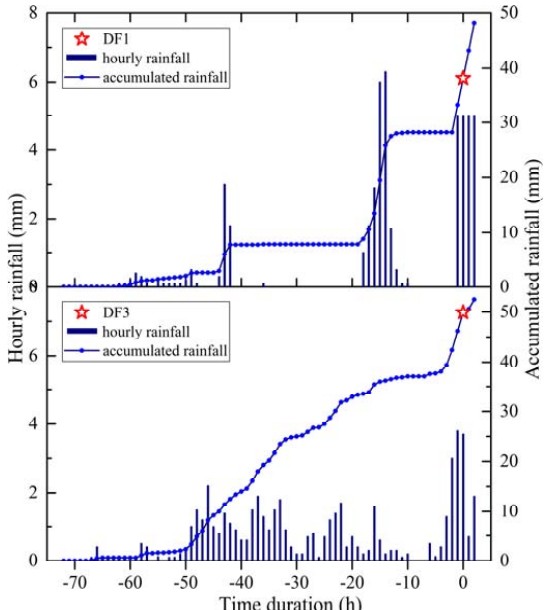


Figure 9 Variation of the rainfall accumulation prior to DF1 and DF3 (no rainfall before DF2)

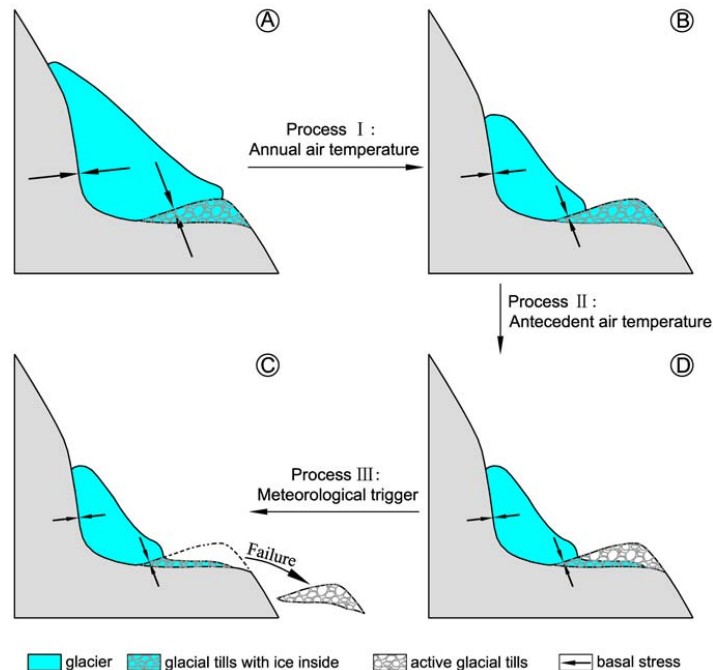


Figure 10 Changes in a glacier and frozen glacial till before periglacial debris flow initiation( A:
glacial covered glacial tills; B: uncovered and frozen glacial tills; C: active glacial tills; D: failure

of glacial tills)