# Peer review of "Meteorological factors driving glacial till variation and the associated"

_Natural Hazards and Earth System Sciences, 2016_

## Referee Comment (RC1) · F. Comiti (Referee) · 24 Oct 2016

The ms presents an interesting study on debris flow acitivity in a poorly investigated area, where climate warming is likely affecting slope stability. The value of the ms in my opinion lies on describing how several factors relevant for debris flow initiation are at play in high elevations, periglacial areas. I am strongly convinced that the results presented here for the Tibetan Plateau are indeed valid also for other geographical regions worldwide. Nonetheless, I think the work is not ready yet fo publication, as the text requiires some ameliorations in the following points (numbers represent ms lines):

- English must be polished and ameliorated by a native speaker - Title: I suggest "mophological factors driving glacial till variations and.."

Introduction - 26: permafrost degradation instead of retreat - 34: not clear the expression "traffic/drainage" - 38-40: not only also in the Andes and elsewhere ! please organize this phrase to be more comprehensinve - 62: "lake" after glacier is probably missing - 69: not clear what is the meaning of "the perfect object". please rephrase

Background - 94-96: not clear why the river channel shading should be relevant for glaciers ? - 114-115: please remove, very poetic but non scientific sentence - 152: Since when has the station been operated ? - 159-162: not convincing. please explain better

Analysis and results

- 176: mean instead of overall ? - 179-184: in all this paragraph one wonders the role of snow vs rainfall in the measurements. Is snowfall measured ? How is it relevant ? - 184: what is normal ? - 188 and 195: I don't undestand why you say tha rainfall incresed at line 195 whereas before you said it was reduced. Please check. I am not sure about the hot-dry and hot-wet. In one case it was not so hot, you report - 208 and 211: please describe what is SPOT and TM - 217: from 2000 - 219-220: values better expressed in hectares - 225: equation is not needed, it is just a simple relative variation ratio - 246: not clear, why the increase may have contributed to glacier retreat ? Please check or rephrase - 275: how many are several ? not precise ! - 284-287: but DF1 was a much larger event compared to the others. Not sure about this interpretation. - 300: why 5 mm/hr ? not clear - 323: sediment instead of soil mass - 336: scope of soil source ? not clear - 336: periglacial - 348: gravitation ? do you mean weight ? better to talk about pressures - 358: depth instead of coverage - 360: not clear "at the junction with the slope" - 360-363: this entire phrase is not clear at all. please rewrite it. Indeed, also Figure 10 (especially C and D) is not clear, and more detailed description should be provided in the caption - 383: scope ? - 384: internal mass of what ? -

391: stepwise manner ? not clear - 406: glacier limited ? maybe till is missing - 412: rainfall related to air temperature fluxes ? this is obvious - 423: is it possible that small events (is failure meaning debris flows? if so should be changed) had cleared the entire source area from active till ? - 438: first year of what ? - 440-441: not always unlimited. not clear why its activity depends on glacier retreat - 443-445: the four phases are quite obvious and could be skipped - 455: available scientific literature Figure 5: in the caption "Mean annual air temperature"

Best wishes
* * *

---

## Author Comment (AC1) · 13 Nov 2016

[revised manuscript text omitted]

批注 [A11]: comment to review #1: mean instead of overall

批注 [A12]: comment to review #1: in all this paragraph one wonders the role of snow vs rainfall in the measurements. Is snowfall measured ? How is it relevant ? -It's a pity that we cannot collect the data on snow so the impact of snow cannot be analyzed.

批注 [admin13]: comment to review #1: what is normal ?

批注 [A14]: comment to review #1: I don't undestand why you say tha rainfall incresed at line 195 whereas before you said it was reduced. In this paper, the year before the three debris flows is hot comparing with the year before.

**(2) Changing of glacier in Tianmo valley**

[revised manuscript text omitted]

批注 [A17]: comment to reviewer 1#: equation is not needed, it is just a simple relative variation ratio

批注 [A18]: comment to reviewer 1#: not clear, why the increase may have ontributed to glacier retreat ? Please check or rephrase -As there is no data on snow, it is just a speculation.

temperature; that is the lowest temperature, the mean temperature and daily rainfall from June to September in 2007 and 2010 (Figure 8).

According to Figure 8, the lowest air temperature was below 0 at the end of June, 2007. At the beginning of July, the air temperature started to rise quickly which continued until early September when DF1 occurred, this demonstrates that the high air temperature in July and August contributed to DF1.

According to Figure 8, the air temperature was high from early July to late August, and another high air temperature period emerged in early September. When DF2 occurred in late July the air temperature had reached the maximum for that year, which suggests that the air temperature in early and middle July was responsible for DF2. After DF2 occurred, the air temperature in August began to prepare for DF3.

Antecedent air temperature fluctuation includes the air temperature and its duration. The air temperature and duration before debris flows are variable, making them difficult to evaluate. The accumulation of positive air temperature is usually applied to analyze the impact of air temperature on glacier melting (Rango and Martinec, 1995), which can be expressed as:

$$T_{PT} = \sum_{i=-n}^{0} T_i(T_i > 0) \qquad (21)$$

Where $T_{PT}$ is the positive air temperature accumulation, °C and $T_i$ is the average daily air temperature; only $T_i > 0$ is included.

Because air temperature is successive, it is difficult to determine the beginning of positive air temperature accumulation. Glacial tills can lessen the heat that penetrates into them, and the low air temperature can only contribute to the upper thin layer; moreover, freeze-thaw cycles exist when the lowest air temperature is less than 0°C. From this point of view, the beginning of positive air temperature accumulation is defined as the time at which the lowest air temperature exceeds 0°C for two or three several successive days or the last debris flow.

批注 [A19]: comment to review #1: how many are several ? not precise !

Based on the above method, we can deduce that the positive air temperature accumulation began when the lowest air temperature exceeded 0°C for several successive days, starting on June 28th, 2007 and June 9th, 2010 corresponding to DF1 and DF2, respectively, and on July 26th, 2010 for DF3, following DF2. The duration and $T_{PT}$ were calculated for each debris flow event , the result was 69 days and 517.9°C, 47 days and 332.1°C, 42 days and 320.4°C (Figure 8) for DF1, DF2, and DF3, respectively. The result showed that $T_{PT}$ for DF1 is much larger than the other two, and  which is coincidence to the fact that there was no debris flows in the past dozens of years and  extraordinary external forces such as larger $T_{PT}$ is required to  destroyed the long-term balance.

**(4) Triggering conditions**

The continuous nature of the air temperature limits the possibility for debris flows triggered by a sole abrupt increase in air temperature; and since the previous air temperature trend cannot be neglected, it is of no sense to study air temperature triggers.

Antecedent rainfall is a factor that favours debris flows. In our analysis, the rainfall over the three days preceding a debris flow event is given in Figure 9.

Before DF1, the air temperature was high, and continued through July and August. The $T_{PT}$ reached 517.9°C. According to the local forest guard, an isolated convective storm occurred prior to DF1 though no rainfall was recorded at the Bomi meteorological station or in the downstream area at that time. In Figure 9, as the rainfall right before DF1 occurred was not recorded by Bomi metrological station, we added to the rainfall intensity (about 5 mm/h according to the description of the forest guard)  before DF1 to account for the storm, which does not reflect the rainfall during storm conditions. We can therefore conclude that this isolated convective storm initiated DF1, while the long-term high air temperature trend had paved the road for DF1. Considering a large deglaciated area, several other periglacial debris flows simultaneously also occurred near Tianmo Valley (Deng et al,

批注 [A20]: comment to review #1: but DF1 was a much larger event compared to the others. Not sure about this nterpretation.

批注 [admin21]: comment to review #1: why 5 mm/hr ? not clear

[revised manuscript text omitted]

---

## Referee Comment (RC2) · Anonymous Referee #2 · 14 Nov 2016

**RE:** NHESS 2016 251        Deng et al.  Meteorological factors driven glacial till changing and the associated periglacial debris flows in Tianmo Valley, southeast Tibetan Plateau

**Overview**

This paper tries to find a correspondence between physically based triggering factors and three occurred debris flows. The english form of the paper is very poor (e.g. at line 32 "located in downstream") and is not acceptable for publication. Moreover, authors should find a better characterization for the peri-glacial debris flows. In other words they could start the differences between runoff generated debris flows and the peri-glacial debris flows. The last issue is that the text is confused: a more synthetic and schematic approach would help the readers. The following are the detailed comments and specifications.

1. The acronyms as TM, DF1, DF2, DF3, SPOT should be defined before their use. Moreover, their use in the abstract should be avoided as much as possible.

2. As runoff generated debris flows, periglacial debris flows are triggered by a water stream that entrain sediments and forms a solid-liquid wave. Water stream is the result of one or a combination of three factors: runoff due to rainfall, melting ice and outburst floods. Due to this similarity between runoff generated debris flow, authors should introduce something more about runoff generated debris flows: the runoff generated debris flows initiate when a peaked runoff hydrograph (Kean et al., 2012; Rengers et al., 2016 and Gregoretti et al., 2016) flow over or impact debris deposits entraining solid material forming the so called debris flow (Berti and Simoni, 2005; Cannon et al., 2008; Coe et al., 2008; Gregoretti and Dalla Fontana, 2008, Theule et al. 2012, Hurlimann et al., 2014, Degetto et al., 2015, Hu et al., 2016).

3. The use of rainfall threshold for explaining the effect of the air temperature should better addressed by explaining that first air temperature increase causes melting and as consequence an abundance of stream water. Therefore, respect to runoff generated debris flows, the rainfall needed for providing the exact critical discharge for debris flow triggering is much minus.

4. Upstream noise (line 116) could be due to slides or rock fall triggered by previous rainfall?

5. Lines 156-163: debris flow are usually triggered by abundant runoff. Abundant runoff is usually provided by convective rainfalls of high intensity and short duration rainfall (Berti

and Simoni, 2005; Gregoretti and Dalla Fontana, 2008). This type of rainfall is characterized by an high spatial variability (Gregoretti et al., 2016). The same authors at line 291 state that a convective storm occurred before DF1 while at the Bomi station no precipitation was recorded. Therefore, please justify in another way the use of the Bomi station. (i.e. data from this station can be used for long-period analysis of cumulative annual rainfall).

6. Lines 186: DF2 and DF3 should defined before their use.

7. Line 208: what is SPOT? Is it the acronym of?

8. Line 228: the writer does not understand the unit measurements for the relative glacial retreat provided by equation (1): what is the duration of year n? According to equation (1) D should be 1/n as dimensions (Area/Area = 1).

9. Lines 300-301: the writer does not understand the meaning of this sentence.

10. Lines 330-340: the sentences seem not clear.

11. Line 421: this statement (no debris flow occurred) contradicts line 128 (debris flows were triggered).

12. There is a clear dependence of debris flow occurrence on the air temperature, while that on rainfall is minus evident (also because direct rainfall measurements are missing in the triggering areas of examined debris flows). This could be explained by the following consideration: debris flow is generated by runoff and runoff is due to the rainfall precipitated upstream the triggering area. This is the mean reason because two debris flows occurred in September. In that month the areas not covered by snow should have reached the largest extension of the year and therefore, runoff in the downstream area should increase. About rainfalls, these could be subjected to an high spatial variability.

Berti, M., and A. Simoni (2005), Experimental evidences and numerical modelling of debris flow initiated by channel runoff, *Landslide*, 2, 171--182.

Cannon, S., Gartner J.E., Wilson, R.C., Bowers, J.C., Laber, J.L. 2008. Storm rainfall conditions for floods and debris flows from recently burned areas in Southwestern Colorado and Southern California. *Geomorphology*, 96, 250-269.

Coe, J.A., Kinner D.A., Godt, J.W., 2008. Initiation conditions for debris flows generated by runoff at Chalk Cliffs, central Colorado. *Geomorphology*, 96, 270-297.

Degetto, M., Gregoretti, C., and Bernard M. 2015. Comparative analysis of the differences between using LiDAR contour-based DEMs for hydrological modeling of runoff generating debris flows in the Dolomites. Frontier in Earth Sciences. 3:21, doi:10.3389/feart.2015.00021

Gregoretti, C., Dalla Fontana G., 2008. The triggering of debris flow due to channel-bed failure debris flow in some alpine headwater basins of the Dolomites: analyses of critical runoff. *Hydrological Processes*. 22, 2248-2263.

Gregoretti C., Degetto M., Bernard M., Crucil, G., Pimazzoni A., De Vido G., Berti M., Simoni  A. Lanzoni S. Runoff of small rocky headwater catchments: Field observations and hydrological modeling. *Water Resources Research*. 52(8) doi: 10.1002/2016WR018675

Hu, W., Dong, X.J., Wang, G.H., van Asch T.W.J. and Hicher P.Y. 2016. Initiation processes for run-off generated debris flows in the Wenchuan earthquake area of China, Geomorphology. 253, 468-477. http://dx.doi.org/10.1016/j.geomorph.2015.10.024

Hurlimann M., Abanco C., Moya, J., Vilajosana I. (2014). Results and experiences gathered at the Rebaixader debris-flow monitoring site, Central Pyrenees, Spain.  *Landslides*. doi:10.1007/s10346-013-0452-y 161-175

Kean J.W., Staley D.M., Leeper R.J., Schmidt K.M., Gartner J.E. (2012). A low-cost method to measure the timing of postfire flash floods and debris flows relative to rainfall. *Water Resources Research*, 48, W05516, doi:10.1029/2011WR011460

Rengers, F.K., L.A. McGuire, J.W. Kean and D.E. Hobley (2016), Model simnulations of flood and debris flow timing in steep cachments after wildfire, *Water Resources Research*, 52, doi:10.1029/2015WR018176.

Theule, J.I., Liebault, F., Loye, A., Laigle, D., and Jaboyedoff, M., 2012. Sediment budget monitoring of debris flow and bedload transport in the Manival Torrent, SE France. *Natural Hazard Earth Sciences*, 12, 731-749.

.

.

---

## Author Response (AR1)

**Answers to reviewer #1:**

- English must be polished and ameliorated by a native speaker - Title: I suggest "mophological factors driving glacial till variations and.."

Answers: we've changed the title and the English will be polished by a native speaker.

Introduction -

  26: permafrost degradation instead of retreat -

Answers:We've made the change in the manuscript. See line 28.

  34: not clear the expression "traffic/drainage" -

Answers:We've made the change in the manuscript. See line 38.

38-40: not only also in the Andes and elsewhere ! please organize this phrase to be more comprehensinve

Answers:We've made the change in the manuscript. See line 43.

- 62: "lake" after glacier is probably missing

Answers:We've made the change in the manuscript. See line 67.

- 69: not clear what is the meaning of "the perfect object". please rephrase Background

Answers:We've deleted this in the manuscript. See line 74.

In Chinese we all see the perfect object or perfect way to show this is a wonderful case.

- 94-96: not clear why the river channel shading should be relevant for glaciers ?

Answers:The river channel in the watershed is sheltered by shade and not directly affected by sunlight, resulting in less solar radiation and a location at which a small trough glacier can form.

- 114-115: please remove, very poetic but non scientific sentence

Answers:We've deleted this in the manuscript. See line 120.

- 152: Since when has the station been operated ?

Answers:It was built since 1955. See line 161.

- 159-162: not convincing. please explain better

Answers:We've changed this in the manuscript. See line 167~172.

Analysis and results

- 176: mean instead of overall ?

Answers:We've changed this in the manuscript. See line 187.

- 179-184: in all this paragraph one wonders the role of snow vs rainfall in the measurements. Is snowfall measured ? How is it relevant?

Answers:It's a pity that the data on snow is not available so the impact of snow cannot be analyzed.

- 184: what is normal ?

Answers:We've changed this in the manuscript. It means the mean rainfall state See line 195.

- 188 and 195: I don't understand why you say the rainfall increased at line 195 whereas before you said it was reduced. Please check. I am not sure about the hot-dry and hot-wet. In one case it was not so hot, you report

Answers:We've changed this in the manuscript. See line 206.

It's our mistake and we've changed this. Besides we did not study of the weather condition if hot-dry and hot-wet is favorable for debris flows, instead, this is a conclusion by Lu and Li(1989) .

Lu, R. R., and Li, D. J.: Ice-Snow-Melt Debris Flows in the Dongru Longba, Bomi county, Xizang. J Glac Geocry, 11(2), 148-160, 1989. (In Chinese)

- 211: please describe what is SPOT and TM

Answers:We've made the change in the manuscript. See line 222~225.

TM image is taken by the No. 4 or 5 thematic mapper carried on the satellite Landsat which is belong to the USA.

Satellite SPOT is short for Systeme Probatoire d'Observation de la Terre belong to France. the SPOT image is taken by this satellite.

- 217: from 2000 - 219-220: values better expressed in hectares

Answers:In this manuscript we did not use hectares considering that it may not useful for the changing of the eastern branch of glacier; instead, the relative glacier retreat rate is applied to show the changing rate.

- 225: equation is not needed, it is just a simple relative variation ratio

Answers:We've delete equation 1.

- 246: not clear, why the increase may have contributed to glacier retreat ? Please check or rephrase

Answers:after deep consideration, we decide to delete this. See line 259.

- 275: how many are several ? not precise !

Answers:We've checked and made some changes. See line 292~293.

Considering the frozen and thawing cycle in the source area, and one day with air temperature larger than 0 could be an exception so we defined the beginning since two or three successive days over 0 ℃ and none is lower than 0 in the following days.

According to the fluctuation of air temperature, we regards two or three days

- 284-287: but DF1 was a much larger event compared to the others. Not sure about this interpretation.

Answers:We've checked and made some changes. See line 300~307.

- 300: why 5 mm/hr ? not clear

Answers:We've checked and made some changes. See line 320~321.

- 323: sediment instead of soil mass

Answers:We've checked and made some changes. See line 345.

- 336: scope of soil source ? not clear

Answers:We've checked and made some changes and it should be the boundary and which can be applied tu calculate the volume. See line 358.

- 336: periglacial

Answers:We've checked and made some changes. See line 358.

- 348: gravitation ? do you mean weight ? better to talk about pressures

Answers:We've checked and made some changes and it should be glacial pressure. See line 371~372.

- 358: depth instead of coverage

Answers:We've checked and made some changes. See line 381.

- 360: not clear "at the junction with the slope"

Answers:We've checked and made some changes and it should be glacial pressure. See line 380~383.

- 360-363: this entire phrase is not clear at all. please rewrite it. Indeed, also Figure 10 (especially C and D) is not clear, and more detailed description should be provided in the caption

Answers:We've paraphrased line 376~383. Also for C and D and we and some explanation to the failure process according to the comment of reviewer #2. See line 428~453.

- 383: scope ?

Answers:We've checked and made some changes. See line 416.

- 384: internal mass of what ?

Answers:We've checked and made some changes. See line 417.

- 391: stepwise manner ? not clear

Answers:We've checked and made some changes. The surface layer should experienced the failure first and then the layers below. See line 424.

- 406: glacier limited ? maybe till is missing

Answers:We've checked and made some changes. we want to express that the ice melt water in a short time should be limited as glacier is limited. See line 454~457.

- 412: rainfall related to air temperature fluxes ? this is obvious

Answers: here we want to express that the triggering condition of debris flows in Alps is controlled by rainfall first and the air temperature. See line 461~462.

- 423: is it possible that small events (is failure meaning debris flows? if so should be changed) had cleared the entire source area from active till ?

Answers: This is in line 476 and we think the small events cannot clear the entire source area.

--the active till is in high area and when it slide along the slope with large gradient, liquefaction can take place to generate debris flows; of course, some can also deposit in the nearby gully for low energy.

--the small events cannot clear all the active till, or the three large events will be not existed.

- 438: first year of what ?

Answers: We've add more explanation and see it in line 494.

- 440-441: not always unlimited. not clear why its activity depends on glacier retreat

Answers: In the above, four processes has been given. The glacial debuttressing, the process that active glacial till emergence are all influenced by glacier retreat.

- 443-445: the four phases are quite obvious and could be skipped

Answers: It has been deleted. See line 502

- 455: Figure 5: in the caption "Mean annual air temperature"

Answers: It has been changed. See figure 5.

**Answers to Review #2:**

1. The acronyms as TM, DF1, DF2, DF3, SPOT should be defined before their use. Moreover, their use in the abstract should be avoided as much as possible.

Answers:We've made the change in the manuscript. See line 222~225.

TM image is taken by the No. 4 or 5 thematic mapper carried on the satellite Landsat which is belong to the USA.

Satellite SPOT is short for Systeme Probatoire d'Observation de la Terre belong to France. the SPOT image is taken by this satellite.

2. As runoff generated debris flows, periglacial debris flows are triggered by a water stream that entrain sediments and forms a solid-liquid wave. Water stream is the result of one or a combination of three factors: runoff due to rainfall, melting ice and outburst floods. Due to this similarity between runoff generated debris flow, authors should introduce something more about runoff generated debris flows: the runoff generated debris flows initiate when a peaked runoff hydrograph (Kean et al., 2012; Rengers et al., 2016 and Gregoretti et al., 2016) flow over or impact debris deposits entraining solid material forming the so called debris flow (Berti and Simoni, 2005; Cannon et al., 2008; Coe et al., 2008; Gregoretti and Dalla Fontana, 2008, Theule et al. 2012, Hurlimann et al., 2014, Degetto et al., 2015, Hu et al., 2016).

Answers: We've made the change in the manuscript. See line 428~453.

Another kind of failure can take place by the increased water stream that entrain sediments and forms a solid-liquid wave if the channel is charged with loose ravel. This kind of water stream could be the combination of the three factors, including rainfall, melting ice or the overflow when the glacier collapse falling down into the downwards water pool. The runoff can generate debris flows when a peaked runoff flow over debris deposits(Kean et al., 2012; Gregoretti et al., 2016 ) and pose hydrodynamic forces acting on the surface elements of the debris layer(Tognacca et al. 2000, Gregoretti ,2000; 2005). The concentration of runoff in the channel bottom causes erosion of the debris surface layer and then extends to the layers below with whole or partial mobilization of the bed material. The inclusion of bed material in the water stream generates debris flow (Gregoretti, 2008).

3. The use of rainfall threshold for explaining the effect of the air temperature should better addressed by explaining that first air temperature increase causes melting and as consequence an abundance of stream water. Therefore, respect to runoff generated debris flows, the rainfall needed for providing the exact critical discharge for debris flow triggering is much minus.

Answers: We've made the change in the manuscript. See line 462~474.

The portion of rainfall and air temperature required for debris flows triggering could be negative. Air temperature increase causes melting and water runoff, and the rainfall needed for providing the percolating flows or exact critical discharge for debris flow triggering would be much less. Beside, the required rainfall, like the intensity and duration, may also require other preconditions, such as the distribution of glaciers and frozen glacial tills and the terrain of the source area

4. Upstream noise (line 116) could be due to slides or rock fall triggered by previous rainfall?
Answers:

This noise should be come from the glacier collapse or slide while not the initiate of debris flows.

5. Lines 156-163: debris flow are usually triggered by abundant runoff. Abundant runoff is usually provided by convective rainfalls of high intensity and short duration rainfall (Berti and Simoni, 2005; Gregoretti and Dalla Fontana, 2008). This type of rainfall is characterized by an high spatial variability (Gregoretti et al., 2016). The same authors at line 291 state that a convective storm occurred before DF1 while at the Bomi station no precipitation was recorded. Therefore, please justify in another way the use of the Bomi station. (i.e. data from this station can be used for long-period analysis of cumulative annual rainfall).
Answers: We've made the change in the manuscript. See line 167~172.

According to our statistics on rainfall data in the area, the rainfall often enjoys the similar intensity for the long-term rainfall process from Guxiang to Songzong which means the there is no large rainfall gradient between Tianmo valley and Bomi meteorological station; however, for the convective rainfall process, rainfall can take place in a small area. In the manuscript, we try to combined the rainfall data in Bomi and the memory of the local citizen to make sure the rainfall process.

6. Lines 186: DF2 and DF3 should defined before their use.

Answers:

DF1, DF2 and DF3 has been debris defined in the part of disaster history.

7. Line 208: what is SPOT? Is it the acronym of?

Answers:We've made the change in the manuscript. See line 222~225.

Satellite SPOT is short for Systeme Probatoire d'Observation de la Terre belong to France. the SPOT image is taken by this satellite. We've add this in the manuscript.

8. Line 228: the writer does not understand the unit measurements for the relative glacial retreat provided by equation (1): what is the duration of year n? According to equation (1) D should be 1/n as dimensions (Area/Area = 1).

Answers:We've made the change in the manuscript. See line 240~242.

 'n' in the equation stands for the time interval of the TM image series, and n=3.

the relative glacial retreat means the annual glacier retreat for each square kilometers.

According to reviewer 1#, we've delete this equation (l).

9. Lines 300-301: the writer does not understand the meaning of this sentence.

Answers:We've made the change in the manuscript. See line 320~321.

In Figure 9, as the rainfall right before DF1 occurred was not recorded by Bomi metrological station, we added to the rainfall intensity (about 5 mm/h according to the description of the forest guard) before DF1 to account for the storm, which might    not reflect the real rainfall process.

10. Lines 330-340: the sentences seem not clear.

Answers:We've made the change in the manuscript. See line 355~362.

We want to make the point that if the water runoff controls the magnitude and frequency of debris flows, larger debris flows could occur before when the larger rainfall process took place. Instead, debris deposits that rainfall or ice melt water can trigger or the water runoff can entrain is regarded as the perquisite.

11. Line 421: this statement (no debris flow occurred) contradicts line 130 (debris flows were triggered).

Answers:We've made the change in the manuscript.

We might not say this clearly that make the reviews difficult to understand.

Line 421 says 'DF1 occurred at the end of a prolonged period of high air temperature, prior to this, there were instances of failure but no large-scale debris flows.' And in line 130, this debris flows(DF2) occurred after DF1 while not prior to DF1.

12. There is a clear dependence of debris flow occurrence on the air temperature, while that on rainfall is minus evident (also because direct rainfall measurements are missing in the triggering areas of examined debris flows). This could be explained by the following consideration: debris flow is generated by runoff and runoff is due to the rainfall precipitated upstream the triggering area. This is the mean reason because two debris flows occurred in September. In that month the areas not covered by snow should have reached the largest extension of the year and therefore, runoff in the downstream area should increase. About rainfalls, these could be subjected to an high spatial variability.

Answers:We've made the change in the manuscript. See line 502~507.

Thanks for the comment and we've added some of them in the discussion and conclusion.

[revised manuscript text omitted]

---

## Referee Report (RR1)

**RE:** NHESS 2016 251        Deng et al. Meteorological factors driven glacial till changing and the associated periglacial debris flows in Tianmo Valley, southeast Tibetan Plateau

**Overview**

Writer opinion is that this work is not yet ready for publication. Main deficiencies are the poor english and some confusion in presenting data, analysis and results. A more schematic approach is required, sometimes, it seems a shopping list. I encourage the authors to a careful review of their work. The help of a native english speaker or of a collegue with fluent english will improve the review.

The detailed explanations regarding these points and other spotted errors are as follows:

**Abstract**

Line 14 The writer does not understand if there were three single events of debris flows or each event was characterized by several debris flows in a same area. According to line 135 it seems that DF2 was not a single event but more events in the same time. Authors should clearly distinguish it.

Line 18  spikedm?? What is it?

Line 20  Triggers of periglacial debris flows are multiplied....What does it mean?

Line 21  as in the first and third debris flow is better than as in the first debris flows and third debris flows

**Introduction**

Line 31   that will be expelled out... what does it mean?   debris material provided by glacier........moraine collapse will be expelled out

Line 39 occurs without s  and are instead of is

Line 47  for the case is rarely to be read. What does it mean?

Line 62  the small rainfall threshold  which is the reference of this?

Lines 69-71  Confused text

**Background**

Line 125-128 Sentence a bit confused.

Line 128, 135 and 143: These debris flows or this debris flow?

Line 165  please substitute can with could

**Analysis and results**

Line 181    0.033°C/y instead of 0.033°C/a

Line 237-238  Confused sentence

Line 245  field instead of file

Line 263 confused sentence

Line 265, 272, 291 and 293  air temperature increase?

**Discussion**

Line 325  it should be the triggering factors of the three debris flows, were.....

Lines 340-342  Confused  sentence

Line 362-364  Confused  sentence

Line 365 ..there were no debris flows of large magnitude...  explain why please

Line 393  retrogressive manner: please explain

Line 413-416   Please consider: Runoff can generate debris flow when a peaked flow impacts a debris deposit (Kean et al., 2013; Gregoretti et al., 2016) and entrain sediments due to the hydrodynamic forces it exert on its surface (Tognacca et al., 2000; Gregoretti, 2008).

Line 420-426  Delete the sentence at line 420-423 and write Therefore, runoff provided by rainfall, seepage flow and melting ice or glacier collapse can initiate debris flow with the same mechanism of the runoff generated debris flows in non-glacier areas (Iverson et al., 1997,      Kean et al., 2012).

Please note that the reference of Gregoretti (2008) is missing and that at line 426 it should be Gregoretti and Dalla Fontana (2008).

Line 429  could be negative...please explain

**Conclusions**

Lines 456-458  Confused sentence

Lines 463-464  Confused sentence

References

Line 524  the reference Gregoretti C. Fontana G.D. is not correct. It is Gregoretti, C., Dalla Fontana G., 2008.

Line 526 Please separate the reference of Gruber and Haeberli

**Authors should distinguish triggering factors from the triggering mechanisms. Rainfall, air temperature and ice melting flow are triggering factors. Triggering mechanisms are seepage flow that leads to a landslide failure or runoff (water stream) that entrain sediments forming a solid-liquid current.**

At Figure 2 it seems that glacial till are in a channalized path. This fact stands for runoff generated debris flows. Debris flows initiate when a glacial till failure provides sediments to the channelized path that are entrained by a water stream.

I suggest the authors a better description of triggering mechanism based on photo of the glacial tills. It could add value to the paper.

---

## Referee Report (RR3)

**RE:** NHESS 2016 251R3        Deng et al.  Meteorological factors driven glacial till changing and the associated periglacial debris flows in Tianmo Valley, southeast Tibetan Plateau

**Overview**

The reviewers comments improved the paper that in my opinion is, unfortunately not yet ready for publication. There are still some confusion in presenting data and processes, some unclear sentences, as well the English form of some parts. The following are the detailed comments and specifications.

1. The acronym TM (line 16) should be defined before its use. Moreover, its use in the abstract should be avoided as much as possible.

2. Line 17 the nearby??????

3. Sentence lines 18-20: the glacial till change is a pre-requisite for debris flow formation because it provides sediments. Therefore, the beginning word, "Moreover", can be omitted, and a brief explanation should be introduced.

4. Line 26: "changes,and": the missing of a space after points and commas is seen in all the paper. Please correct.

5. Line 32: "Debris flowsin": the missing of space between two words is seen in all the paper. Please correct.

6. Line 69; maybe "possibility" is better than "complexity".

7. Lines 118-121. "On the morning" is not coherent with "18.00" and add t to even (line 119); I suggest to rewrite the sentence as: ……the triggering area was hit by a rainfall event and after that some loud noise were heard about 18:00…….

8. Lines 121-123  Rewrite the sentences as: a debris flow occurred after a second rainfall event that began at 19:00

9. Lines 122, 131 and 138 debris flows?  The writer does not understand if for a debris flow event the authors intend different debris flows or a debris flow composed by several waves or something else. Please explain in the paper.

10. Line 128: Table 2 is not necessary because it deals with debris flows that are not object of present work.

11. Lines 142-143: the finding of Chen (1991) could be due to the increase of melting water, while in present case debris flow has been triggered by rain storm.

12. Lines 185-189. The sentences could be substituted by In the periods 2000-2004 and 2007 rainfall precipitated from July to September was about 50% of the total annual rainfall while in the periods 2005-2006 and 2008-????was about 32% and ???? respectively.

13. Lines 190-196: a more schematic and concise presentation of data is required.

14. Line 216: mapperson??????

15. Line 261: explain in the caption of Figure 8 the meaning of PT.

16. Line 291: were instead of "was"

17. Lines 295-297. The writer does not understand the first sentence and therefore its link and the sense of the second sentence.

18. Lines 301, 313 and 321: 517.9°C: this high value (in centrigade) is not possible.

19. Line 315-316: from that there had been no…….. the sentence becomes unclear.

20. Line 322 What does it mean a steady rainfall?

21. Line 332 write were after "there"

22. Line 335 and following: write sediment source instead of "soil source"

23. Figure 10: perhaps panels C and D should be inverted.

24. Lines 356-368. The link between the thaw process within the till, its duration and the absence of debris flow in 2006 and 2009 is ill explained or missing.

25. Line 409   in which care????

26. Line 411 Delete "a" before "peaked" and write "peaked runoff flows (Kean et al., 2012, Rengers et al., 2016, Gregoretti et al., 20016) "

27. Line 414 add the references Theule et al., 2012, Hurlimann et al., 2014 and Degetto et al., 2015.

28. Line 415 delete the reference Armanini and Gregoretti, 2005 and Kean et al., 2013.

29. Line 418-421 Rewrite the sentence as: Mechanism of this process lies in the hydrodynamics forces exerted on the surface elements of debris layers and surpassing sediment resistance (Gregoretti, 2008; Recking et al., 2009; Prancevic et al., 2014).

30. Line 425 rewrite the initial part of the sentence as: Therefore, debris flows initiated by landslide failure caused by seepage flow and by channelized runoff….

31. Lines 442-452. These sentences should be resumed in a more concise form. The link of the amount of rainfalls and the triggering of debris flow should be clearly explained. Moreover, authors should consider that the two debris flows triggered by rainfall (DF1 and DF2) when the areas not covered by glacier should have reached the largest extension of the year and therefore, runoff in the downstream area should increase.

32. Line 466-467: what is the meaning of variable air temperature condition? The sentence is unclear.

33. Lines 475-477 Last part of the sentence is not well linked to the previous part.

Degetto, M., Gregoretti, C., and Bernard M. 2015. Comparative analysis of the differences between using LiDAR contour-based DEMs for hydrological modeling of runoff generating debris flows in the Dolomites. Frontier in Earth Sciences. 3:21, doi:10.3389/feart.2015.00021

Gregoretti C., Degetto M., Bernard M., Crucil, G., Pimazzoni A., De Vido G., Berti M., Simoni  A. Lanzoni S. Runoff of small rocky headwater catchments: Field observations and hydrological modeling. *Water Resources Research*. 52(8) doi: 10.1002/2016WR018675

Hurlimann M., Abanco C., Moya, J., Vilajosana I. (2014). Results and experiences gathered at the Rebaixader debris-flow monitoring site, Central Pyrenees, Spain.  *Landslides*. doi:10.1007/s10346-013-0452-y 161-175

 Kean J.W., Staley D.M., Leeper R.J., Schmidt K.M., Gartner J.E. (2012). A low-cost method to measure the timing of postfire flash floods and debris flows relative to rainfall. *Water Resources Research*, 48, W05516, doi:10.1029/2011WR011460

Recking A. 2009. Theoretical development on the effect of changing flow hydraulics on incipient bed load  motion. Water Resources Research. 45, W04401; doi:10.1029/2008WR006826

Rengers, F.K., L.A. McGuire, J.W. Kean and D.E. Hobley (2016), Model simnulations of flood and debris flow timing in steep cachments after wildfire, *Water Resources Research*, 52, doi:10.1029/2015WR018176.

 Theule, J.I., Liebault, F., Loye, A., Laigle, D., and Jaboyedoff, M., 2012. Sediment budget monitoring of debris flow and bedload transport in the Manival Torrent, SE France. *Natural Hazard Earth Sciences*, 12, 731-749.

.

.

---

## Author Response (AR2)

**Meteorological factors driving glacial till variation and the associated periglacial periglacial debris flows in Tianmo Valley, southeast Tibetan Plateau**

M. F. Deng[1,2], N. S. Chen[1]*, and M. Liu[1,2]

([1]Key Laboratory of Mountain Hazards and Surface Process, Institute of Mountain Hazards and Environment, Chinese Academy of Sciences, Chengdu 610041, China;

[2] University of Chinese Academic of Sciences, Beijing 100049, China)

Abstract: Meteorological studies have indicated that high aAlpine environmentss are strongly affected by climate warming, and .pPeriglacial debris flows are frequent in deglaciated regions. The combination of rainfall and air temperature controls the initiation of periglacial debris flows,; and the addition of melt-water due to higher air temperatures enhances the complexity of the triggering mechanism compared to that of storm-induced debris flows. In On the south-eastern Tibetan Plateau, where temperate glaciers are widely distributed, numerous periglacial debris flows have occurred in over the past 100 years, but none had happenedoccurred in the Tianmo watershed until 2007. In 2007 and 2010, three large-scale debris flows occurred in the TianmowatershedValley. In this researchstudy, these three debris flow events were chosen to analyze analyse the impacts of the annual meteorological conditions, including: the antecedent air temperature and meteorological triggers. TM images and field measurements of thenearby glacier suggested that a sharp glacier retreats had existedoccurred in the previous one or to two years preceding the events, which coincided with the spikes indmthe mean annual air temperature. Besides,. Moreover, changing of glacial till changess driven by a prolonged increase in the air temperature is are the a prerequisite of periglacial debris flows. Different factors can trigger The tTriggers of periglacial debris flows are multiplied often coupled, and they could bemay include high intensity rainfall, as in the first debris flows and the third debris flows, or continuous, long-term rising increases in air temperature,s as in the second debris flow events.

Key words: glacial till variation; meteorological factors; periglacial debris flows; southeast Tibetan Plateau

**1. Introduction**

The Alpinealpine environments are strongly vulnerable to climate changes,of which theand alpine glaciers and permafrost are the most sensitive in the form of glacier and permafrostto degradation (Harris et al., 2009; IPCC, 2013). Glacier and

批注 [A1]: Answer to reviewer #2: the three debris flows is of large magnitude while debris flows with small magnitude occurred followed each of the three debris flows; while debris flow with small magnitude is hard to be determined compared with the larger ones.

批注 [A2]: Answer to reviewer #2: Line 18 spikedm?? What is it?

批注 [A3]: Answer to reviewer #2: Triggers of periglacial debris flows are multiplied….What does it mean?

批注 [A4]: Answer to reviewer #2: as in the first and third debris flow is better than as in the first debris flows and third debris flows

[revised manuscript text omitted]

**批注 [A23]:** Answer to reviewer #1: occurs ?

**批注 [A24]:** Answer to reviewer #1: similarly to above, not really the correct verb

**批注 [A25]:** Answers to reviewer #2: Confused sentence

**批注 [A26]:** Answers to reviewer #2: ..there were no debris flows of large magnitude... explain why—the newly formed glacial till is of high shearing strength and low activity, leading to the low possibility of shallow landslide occurring.

(Figure 10 B to C). Because  heat conduction in glacial till is  relatively slow, this process may last for a very long time and  require a high antecedent air temperature.

Heat conduction via the percolation of rainfall and ice melt-water can amplify the depth of active  glacial till (Gruber and Haeberli, 2007), whereas covering the surface glacial till can hinder  a heat flux from penetrating into the deep layers(Noetzli et al., 2007). At a low air temperature, the heat flux should be constrained to the surface layer, and a large heat gradient due to a high air temperature would contribute much more to the heat flux and ice melt in the deep mass. Thus,  the long-term effect of a high air temperature can amplify the active glacial till (Noetzli et al., 2007; Åkerman and Johansson, 2008), under which lies frozen glacial till with a high ice content. The activity of glacial till  varies with depth from high  at the surface  to low in the deep layers, and landslide failure can take place on glacial till slopes in a retrogressive manner, coinciding with long-term air temperature fluctuations,  as active  glacial till is relatively limited in  deglaciated areas.

**(3) Failure of glacial tills**

 Different factors can lead to glacial till failure. Active glacial till slopes with low strength are usually vulnerable, and their failure can occur when the air temperature is above 0 °C (Arenson and Springman, 2005). Rainfall or ice melt water induced by air temperature can  trigger the failure (Figure 10 C to D). This  type of event is called  a shallow landslide , and the failure mechanism lies in the ablation of internal ice particles and the percolation of melt-water, which can initially decrease the soil strength  (Arenson and Springman, 2005; Decaulne et al., 2005). Later, the subsequent rapid percolation of ice melt-water or heavy rainfall can saturate the  debris, decrease soil suction and shearing strength, and form seepage flows that

批注 [A27]: Answers to reviewer #2: retrogressive manner means failure will take place at the surface active layer, followed by the deeper layer with increase of its activity

批注 [A28]: Answers to reviewer #1: what do you mean ? "different factors can lead to glacial till failure" ?

批注 [A29]: Answers to reviewer #2: Authors should distinguish triggering factors from the triggering mechanisms.

canmatrix.Tthe glacial till decrease soil suction and shearing strengthof the glacial till decrease and  then trigger the shallow landslide failure (Springman et al., 2003; Decaulne and Sæmundsson, 2007; Chiarle et al, ., 2007). Whether the failure can induce debris flows is still still dependent on the its ability that it canto entrain the debris layer, in which care the ,otherwise, it candebris is deposited and  cas the flow moves throughharge the channel.

Another kind  type of failure canmight take place when a peaked runoff flows over and entrains debris deposits in the charged channel and reach a critical discharge (Berti and Simoni, 2005; Gregoretti and Dalla Fontana, 2008; Kean et al., 2013; Takahashi, 2014), which is more determined by channel bed slope and grain size of debris (Tognaccaet al., 2000; Gregoretti, 2000; Armanini and Gregoretti, 2005; Kean et al., 2013). byThis kind  type of water streamchannelized runoff could be the a combination of the three factors: ,including rainfall, melting ice or the overflow that forms when the a glacier collapses falling down into the downwardsdownward into a water pool. Mechanism of this process lies in the hydrodynamic forces, created by the channelized runoff, (Kean et al., 2013; Gregoretti et al., 2016) and pose createhydrodynamic forces acting that acting on the surface elements of the debris layer and surpassing resistence of the sediment (Tognaccaet al., 2000, ; Gregoretti, 2000; Armanini and Gregoretti, 2005; Prancevic et al., 2014). The concentration of runoff in the channel bottom causes the erosion of the debris surface layer forming a solid-liquid current at first, and then extends to the layers below with whole or partial mobilization and debris flows was generated (Gregoretti and Dalla Fontana, 2008). , resulting in with whole or partial mobilizationof the bed material. The inclusion of bed material inthe flowing waterwater stream generates a debris flow (Gregoretti and Fontana, 2008).

Therefore, debris flows initiated from seepage flow that leads to a landslide failure or channelized runoff that entrain sediments in the periglacial area is similar with the mechanism of debris flows initiation in non-glacier areasThe Ffluctuationsof in air temperature within a specific low range can result into in limited seepage flow. As Because the glacier in is limited to one valley is limited, it is unlikely for that

批注 [A30]: Answer to reviewer 2#: reference of Gregoretti (2008) is missing and that at line 426 it should be Gregoretti and Dalla Fontana (2008).

批注 [A31]: Answer to reviewer 2#: I suggest the authors a better description of triggering mechanism.....

批注 [A32]: Answer to reviewer 2#: reference of Gregoretti (2008) is missing and that at line 426 it should be Gregoretti and Dalla Fontana (2008).

批注 [A33]: Answer to reviewer 2#: reference of Gregoretti (2008) is missing and that at line 426 it should be Gregoretti and Dalla Fontana (2008).

[revised manuscript text omitted]

---

## Author Response (AR3)

[revised manuscript text omitted]

批注 [admin8]: Answer to the comment:
7. Lines 118-121. "On the morning" is not coherent with "18.00" and add t to even (line 119); I suggest to rewrite the sentence as: ……the triggering area was hit by a rainfall event and after that some loud noise were heard about 18:00……. ... [2]

批注 [admin9]: Answer to the comment:
8. Lines 121-123 Rewrite the sentences as: a debris flow occurred after a second rainfall event that began at 19:00 ... [3]

已下移 [1]: This debris flow event is listed as DF1 in this paper.

已移动(插入) [1]

批注 [admin10]: Answer to the comment:
9. Lines 122, 131 and 138 debris flows? The writer does not understand if for a debris flow event the authors intend different debris flows or a debris flow composed by several waves or something else. Please explain in the paper. ... [4]

批注 [admin11]: Answer to the comment: 10. Line 128: Table 2 is not necessary because it deals with debris flows that are not object of present work. ___the other debris flows cases in table 2 is used to show that the metrological condition in these days is favorable for debris flows ... [5]

批注 [admin12]: Answer to the comment: 11. Lines 142-143: the finding of Chen (1991) could be due to the increase of ... [6]

determined.

[revised manuscript text omitted]

Antecedent air temperature fluctuations include the air temperature and the
duration of variations. The air temperatures and durations before debris flows are
variable and difficult to evaluate. The accumulation of positive air temperature is
often used to analyse the effect of air temperature on glacier melting (Rango and
Martinec, 1995) and can be expressed as follows:

$$T_{PT} = \sum_{i=-n}^{0} T_i (T_i > 0) \tag{1}$$

where $T_{PT}$ is positive air temperature accumulation (°C) and $T_i$ is the average daily air temperature (only $T_i > 0$ is included).

批注 [admin15]: answer to comment: 15. Line 261: explain in the caption of Figure 8 the meaning of PT. PT is T$_{PT}$

Because air temperature is successive, it is difficult to determine the beginning of
positive air temperature accumulation. Glacial tills can decrease the heat that
penetrates into them, and the low air temperature is only observed in the upper thin
layer. Moreover, freeze-thaw cycles exist when the lowest air temperature is less than
0°C. From this perspective, the beginning of positive air temperature accumulation is
defined as the time at which the lowest air temperature exceeds 0°C for two or three
successive days or since the last debris flow.

Based on the above method, we can deduce that positive air temperature
accumulation began when the lowest air temperature exceeded 0°C for several
successive days beginning on June 28$^{th}$, 2007, June 9$^{th}$, 2010, and July 26$^{th}$, 2010,
which correspond to DF1, DF2 and DF3, respectively. The duration and $T_{PT}$ were
calculated for each debris flow event. The results were 69 days and 517.9 °C, 47 days
and 332.1 °C and 42 days and 320.4 °C (Figure 8) for DF1, DF2 and DF3, respectively. The results showed that $T_{PT}$ for DF1 was much larger than the other two $T_{PT}$ values, which coincides with the fact that there were no debris flows in the past dozens of years, and extraordinary external forces such as large $T_{PT}$ are required to disrupt the long-term balance.

**(4) Triggering conditions**

Rainfall in a short can trigger debris flows while it cannot be triggered by a sole abrupt increase in air temperature as the continuous and limited nature of air temperature, instead, air temperature of longer term should be included. In our analysis, the rainfall over the three days preceding a debris flow event is given in

Figure 9.

Before DF1, the air temperature was high, which continued through July and

August. Notably, the $T_{PT}$ reached 517.9°C. According to the local forest guard, an isolated convective storm occurred prior to DF1, although no rainfall was recorded at the Bomi meteorological station or in the downstream area at that time. In Figure 9, as the rainfall right before DF1 occurred was not recorded by the Bomi metrological station, we added approximately 5 mm/h of rainfall intensity (according to the description provided by the forest guard) before DF1 to account for the storm, which might not reflect the real rainfall process. We can therefore conclude that this isolated convective storm initiated DF1, while the long-term high air temperature trend paved the way for DF1. Considering a large deglaciated area, several other periglacial debris flows simultaneously occurred near Tianmo Valley (Deng et al., 2013), which suggests the advantageous meteorological conditions for debris flow initiation.

DF2 occurred when the air temperature reached a peak in 2010. The thaw season began in the middle of June, and $T_{PT}$ reached 332.1 °C. On July 24[th], one day before

DF2, the air temperature reached a maximum value for that year. No rainfall event hit this area preceding DF2 according to the record of Bomi meteorological station, and the local citizens also observed no rain. The trigger of DF2 was likely the continuous

批注 [admin16]: answer to comment: 16. Line 291: were instead of "was"

Antecedent rainfall is a factor that favours debris flows.

批注 [admin17]: answer to comment: 17. Lines 295-297. The writer does not understand the first sentence and therefore its link and the sense of the second sentence.

批注 [admin18]: answer to comment: 18. Lines 301, 313 and 321: 517.9°C: this high value (in centrigade) is not possible.
___517.9°C is the positive air temperature accumulation which can be calculated by the eqation (1).

批注 [admin19]: answer to comment : 19. Line 315-316: from that there had been no…….. the sentence becomes unclear.

percolation of meltwater due to the long-term increase in air temperature.

According to field interviews, several debris flows of small magnitude occurred before DF3. The air temperature decreased in late August but increased to another high value before DF3, and the $T_{PT}$ reached 320.4 °C. Rainfall began 2 days prior to DF3 and lasted the entire day before DF3. According to the rainfall trend at the Bomi meteorological station, the rapid increase in rainfall intensity started 4 hours before DF3 and reached 3.8 mm/h, which was responsible for the initiation of DF3.

**4. Discussion**

In this study, we found that the triggering factors of the three debris flows were high air temperature and rainfall for DF1, high air temperature for DF2 and storm for DF3, respectively. When we analysed the dates and triggers of these events, various questions should be settled first: 1) why did debris flows not occur in 2006 or 2009 when deglaciation reached its peak and more ice meltwater was present; 2) why did DF1 and DF3 occur in September when the air temperature and volume of ice meltwater were decreasing; and 3) why there were no large-scale debris flows triggered by previous heavy storms. Based on our results, we believe that the impact of the water source on the magnitude and frequency of debris flows is relatively small, or more debris flows would form during the early larger storm; instead, the sediment source, including the associated magnitude and activity, may be the predominant control, as reported by Jakob et al. (2005), who noted that channel recharge is a prerequisite for debris flows. However, in most situations, we cannot reach the source area to detect the soil source, and high-tech remote sensing can only distinguish the boundary of the soil source. In the periglacial area where glacial till is often covered by glacier or everlasting snow, a change in the soil source would be highly difficult to detect. In this study, we combine the meteorological conditions and literature reports to discuss the likely variations in glacial tills before debris flows.

**(1) Annual variations in glacial till**

Climate warming is a global trend (IPCC, 2013), and the Tibetan Plateau, as the

批注 [admin20]: answer to comment:
20. Line 322 What does it mean a steady rainfall?

批注 [admin21]: answer to comment :
21. Line 332 write were after "there"

批注 [admin22]: answer to comment :
22. Line 335 and following: write sediment source instead of "soil source"

third pole, is no exception to climate change. According to our statistics, the air temperature in Bomi County has increased by 1.5 °C over the past 45 years (1970~2014). Glacier retreat induced by climate warming has been widely accepted, and recent research suggests that the weaker Indian monsoon could be another reason for such retreat (Yao et al., 2012). Glaciers are always located in concave ground areas and cover large volumes of glacial tills. Glacial pressure can generate normal stress vertical to the slope, which can strengthen the slope stability. The effect of glaciers on slope stability is called glacial debuttressing (Cossart et al., 2008). As deglaciation continues, the result could lead to the exposure of the frozen glacial tills (Figure 10 A to B) and smaller glacial debuttressing.

The retreats of glaciers and glacial tills due to climate warming are quite different. The newly formed bare glacial till is frozen with a high ice content. The cohesion of the ice particles creates a bare glacial till with high shearing strength and stability. Deglaciation is accompanied by the melting of internal ice particles, which can greatly enhance the activity. This process first occurred at the surface layer of glacial till, followed the layers below, resulting in enlargement of active debris. As the debris obstruct heat fluxes from penetrating into the layer below, so the melting rate of internal ice particles is quite slower than that of glacial retreat (Takeuchi et al., 2000), result into a strong heat gradient at the surface while limited in deep layers, which means the activity of the debris decline with depth and long term high air temperature is required to enhance the activity in a deeper layer. As the ablation rates is quite low, only the surface layer is highly active and the sediment is relatively limited. Therefore, no debris flows of large magnitude could occurred in 2006 and 2009 when glacier retreat reached a maximum while the active glacial till is restricted to the surface layer.

**(2) Variation in glacial till on antecedent days**

After the long, cold winter, glacial tills become frozen. If a regressive glacier does not recover in the winter, glacial tills are covered by snow. As the air temperature increases again, the surface snow melts first, followed by the internal ice particles.

批注 [admin23]: answer to comment: 24. Lines 356-368. The link between the thaw process within the till, its duration and the absence of debris flow in 2006 and 2009 is ill explained or missing.

[revised manuscript text omitted]

批注 [admin30]: answer to comment: 31. Lines 442-452. These sentences should be resumed in a more concise form. The link of the amount of rainfalls and the triggering of debris flow should be clearly explained. Moreover, authors should consider that the two debris flows triggered by rainfall (DF1 and DF2) when the areas not covered by glacier should have reached the largest extension of the year and therefore, runoff in the downstream area should increase.

Tianmo Valley. Debris flows are generally not initiated in the year when the mean annual air temperature spikes, as the melting of internal ice particles lags behind the rate of glacial retreat resulting from a prolonged increase in air temperature.

Glacial till is unlimited in deglaciated areas, and its activity relies on glacial retreat and internal ice particle melting. Glacial till changes induced by increased air temperature are the first steps in forming periglacial debris flows compared to storm-induced debris flows in non-glacier areas. Glacial tills require a four-phase process prior to debris flow occurrence. In this process, the variation in air temperature drives the glacial till change, including causing glacier recession, producing bare glacial till and enhancing the glacial till activity. Debris flows can occur when a sufficient amount of active glacial till exists and rainfall-induced seepage or runoff is more likely to generate debris flows.

It is difficult to observe glacial till changes in source areas of debris flows, and the analysis of the phase conversion of glacial till in this study is based on the triggering conditions and other literature findings. Indeed, the meteorological conditions, such as the antecedent air temperature and meteorological triggers that drive the phase conversion, are partly coupled and difficult to distinguish.

**Acknowledgements**: This research was supported by the National Natural Science Foundation of China (grant nos. 41190084, 41402283 and 41371038) and the "135" project of IMHE, CAS. We wish to acknowledge the editors of the Natural Hazards and Earth System Science Editorial Office and the anonymous reviewers for their constructive comments, which helped us improve the contents and presentation of the manuscript.